# Emergent layer stacking arrangements in c-axis confined MoTe$_2$

James L. Hart [1], Lopa Bhatt[2], Yanbing Zhu[3], Myung-Geun Han [4], Elisabeth Bianco[5], Shunran Li[6,7], David J. Hynek [7,8], John A. Schneeloch [9], Yu Tao[9], Despina Louca [9], Peijun Guo [6,7], Yimei Zhu [4], Felipe Jornada [10], Evan J. Reed[10], Lena F. Kourkoutis [2,5] & Judy J. Cha[1,11] ✉

The layer stacking order in 2D materials strongly affects functional properties and holds promise for next-generation electronic devices. In bulk, octahedral MoTe$_2$ possesses two stacking arrangements, the ferroelectric Weyl semimetal T$_d$ phase and the higher-order topological insulator 1T' phase. However, in thin flakes of MoTe$_2$, it is unclear if the layer stacking follows the T$_d$, 1T', or an alternative stacking sequence. Here, we use atomic-resolution scanning transmission electron microscopy to directly visualize the MoTe$_2$ layer stacking. In thin flakes, we observe highly disordered stacking, with nanoscale 1T' and T$_d$ domains, as well as alternative stacking arrangements not found in the bulk. We attribute these findings to intrinsic confinement effects on the MoTe$_2$ stacking-dependent free energy. Our results are important for the understanding of exotic physics displayed in MoTe$_2$ flakes. More broadly, this work suggests c-axis confinement as a method to influence layer stacking in other 2D materials.

In layered van der Waals (vdW) solids, exotic quantum phenomena can be engineered via the layer stacking. For instance, the twist angle in bilayer graphene influences the low-energy electronic band structure, allowing for control over magnetic[1], superconducting[2], and topological phases[3]. When the twist angle of a homo-structure is zero, the in-plane displacement between layers, i.e., the layer stacking order, offers an additional control parameter. Examples include emergent ferroelectricity in hexagonal boron nitride[4,5], magnetic order in CrI$_3$[6], and quantum transport in trilayer graphene[7]. In certain cases, the layer stacking order can be dynamically controlled through external stimuli[8–10], which is attractive for device applications. However, our basic understanding of layer stacking energetics, as well as layer-sliding transitions, remains limited.

Octahedrally coordinated MoTe$_2$ and WTe$_2$ are prime candidates for stacking order-dependent devices. In bulk, two stable stacking arrangements exist[11,12]: the low-temperature T$_d$ phase, a ferroelectric Weyl semimetal[13], and the high-temperature 1T' phase, a higher-order topological insulator[14] (Fig. 1a) with a transition temperature (T$_c$) of ~250 K for MoTe$_2$[15] and ~565 K for WTe$_2$[16]. In thin mechanically exfoliated flakes of MoTe$_2$ less than ~20 nm, the temperature-dependent layer stacking transition is suppressed; however, the preferred layer stacking in such flakes is unclear. Raman spectroscopy studies have reached conflicting conclusions, finding either that thin flakes prefer 1T' stacking, or T$_d$ stacking, or alternative stacking sequences distinct from the known bulk phases[17–22]. Electronic transport and quantum oscillation studies suggest that thin flakes adopt T$_d$ stacking, but the

[1]Department of Materials Science and Engineering, Cornell University, Ithaca, USA. [2]School of Applied and Engineering Physics, Cornell University, Ithaca, USA. [3]Department of Applied Physics, Stanford University, Stanford, USA. [4]Condensed Matter Physics and Materials Science Department, Brookhaven National Laboratory, Upton, USA. [5]Kavli Institute at Cornell for Nanoscale Science, Cornell University, Ithaca, USA. [6]Department of Chemical and Environmental Engineering, Yale University, New Haven, USA. [7]Energy Sciences Institute, Yale University, West Haven, USA. [8]Department of Mechanical Engineering and Materials Science, Yale University, New Haven, USA. [9]Department of Physics, University of Virginia, Charlottesville, USA. [10]Department of Materials Science and Engineering, Stanford University, Stanford, USA. [11]Cornell Center for Materials Research, Cornell University, Ithaca, USA. ✉e-mail: jc476@cornell.edu

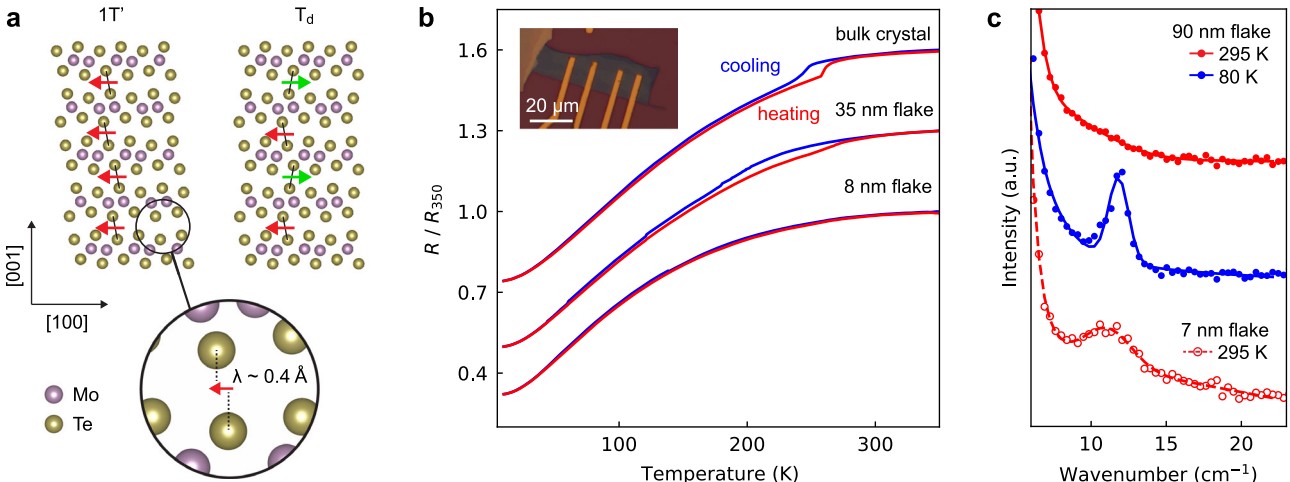

**Fig. 1 | Layer stacking in bulk MoTe₂ and initial characterization. a** Schematics of the 1T′ and $T_d$ phases of MoTe₂ in the *ac*-plane. We use universal coordinates, where the layer-sliding direction is along the *a*-axis for both 1T′ and $T_d$. The overlaid arrows represent the in-plane displacement component of the Te-Te pairs that bridge the vdW gap, here expressed by the vector **λ**. The color of the arrows denotes the displacement direction. **b** Electrical resistance of MoTe₂ of varying thicknesses as a function of temperature, normalized to the resistance at 350 K. Data are offset vertically for clarity (plus 0.3 for 35 nm flake and plus 0.6 for the bulk crystal). Inset shows an optical image of the 8 nm flake. **c** Raman spectroscopy of MoTe₂ flakes as a function of temperature and flake thickness. The spectra show the inter-layer shear mode, which is sensitive to the layer stacking order[19]. Extended Raman data is shown in Supplementary Fig. 1.

evidence is indirect[17,18,23]. In this thickness range, MoTe₂ flakes show a myriad of intriguing phenomena, e.g., enhanced superconductivity[18], superconducting edge currents[24], giant out-of-plane Hall effect[25], helical 1D hinge states[26], and in-plane, third-order nonlinear Hall effect[27]. To fully understand and exploit these behaviors, the stacking order—which dictates the symmetry and topology—must be determined. Moreover, determining the stacking in thin MoTe₂ may serve as a general platform for understanding dimensional effects in other 2D materials. In contrast to MoTe₂, the layer stacking in WTe₂ has not been studied as a function of thickness.

Here, we determine the layer stacking of MoTe₂ and WTe₂ flakes by atomic-resolution scanning transmission electron microscopy (STEM). We find that the layer stacking in thin exfoliated flakes of MoTe₂ does not follow the ordered 1T′ or $T_d$ phases; rather, we observe a number of alternative stacking sequences which lack long-range order. In contrast, the stacking in WTe₂ is well-ordered $T_d$, even for thin flakes. To explain the disordered stacking in MoTe₂, we consider and rule out extrinsic factors such as sample oxidation or interface effects, and we discuss the intrinsic coupling between the flake thickness, stacking arrangement, and free energy. These results are crucial for our interpretation of the various quantum properties exhibited by MoTe₂ flakes, for the future design of MoTe₂-based devices, and for our understanding of thickness-effects in layered 2D materials.

## Results

### Electrical transport and Raman spectroscopy

We initially studied the layer stacking phase transition of MoTe₂ through electrical transport measurements and Raman spectroscopy. Transport measurements of a bulk crystal show a clear thermal hysteresis loop centered at ~250 K, indicative of the expected first-order stacking transition (Fig. 1b)[15]. The thermal hysteresis loop is broadened and partially suppressed for flakes 10 s of nm thick and then fully suppressed for flakes <10 nm. This trend suggests that the stacking transition is mostly quenched in thin exfoliated flakes, consistent with prior reports[17,18,22].

With Raman spectroscopy, the most direct signature of the stacking transition in bulk MoTe₂ is the activation of an inter-layer shear mode at 12 cm⁻¹ (1.5 meV)[19]. This mode is Raman silent in the centrosymmetric 1T′ phase but emerges in the $T_d$ phase owing to inversion symmetry breaking[28,29]. For our measurements of a relatively

thick flake (90 nm), the inter-layer shear mode is absent at room temperature as expected and then activated when measured at 80 K, consistent with the bulk 1T′ to $T_d$ transition (Fig. 1c). Conversely, for a 7 nm thick flake, we unexpectedly observe the shear mode at room temperature (50 K above the bulk $T_c$), though the peak is broadened and softened. This finding is similar to that of ref. 17.

Taken together, our transport and Raman data suggest that for thin flakes, the stacking transition is quenched, and the $T_d$ phase is stabilized up to (at least) room temperature. This interpretation has been advocated in prior reports[17,18]. However, the precise relation between the layer stacking and the electrical resistance is unclear[30,31]. Moreover, the emergence of the inter-layer Raman mode does not guarantee the $T_d$ phase; rather, this mode simply indicates inversion symmetry breaking[28,29]. Alternative stacking sequences could also break inversion symmetry, and for thin flakes, inversion symmetry is necessarily broken at interfaces, even for centrosymmetric crystals. Hence, symmetry-based Raman analysis cannot unequivocally identify the layer stacking, and atomic-scale visualization is needed.

### Room temperature (S)TEM

To directly determine the structure of thin exfoliated MoTe₂ and WTe₂ flakes, we performed high-angle annular dark-field (HAADF) STEM imaging. First, we observe flakes in plan-view (the *ab*-plane) by transferring exfoliated flakes to a STEM grid via a PDMS stamp. In this geometry, the 1T′ and $T_d$ stacking sequences are easily differentiated, as shown with the atomic schematics and STEM image simulations in Fig. 2a. Our experimental imaging of exfoliated MoTe₂ flakes reveals a number of distinct structures, none of which match the pure 1T′ or $T_d$ phases (Fig. 2b). These results indicate that the stacking does not follow either of the bulk phases and that the layer stacking order in thin MoTe₂ flakes is not spatially uniform. In contrast, for exfoliated WTe₂, the observed crystal structure is in excellent agreement with the simulated $T_d$ structure, demonstrating that WTe₂ retains its ordered $T_d$ stacking at room temperature in thin flakes (Fig. 2c).

To better understand the irregular stacking in MoTe₂ flakes, we next studied flakes in cross-section (the *ac*-plane), which allows direct determination of the layer stacking order. As schematically shown in Fig. 1a, the 1T′ and $T_d$ phases can be differentiated based on the Te-Te pairs bridging the vdW gap. For 1T′ stacking, the in-plane displacement component of this pair (the inter-layer shift) is always in the same

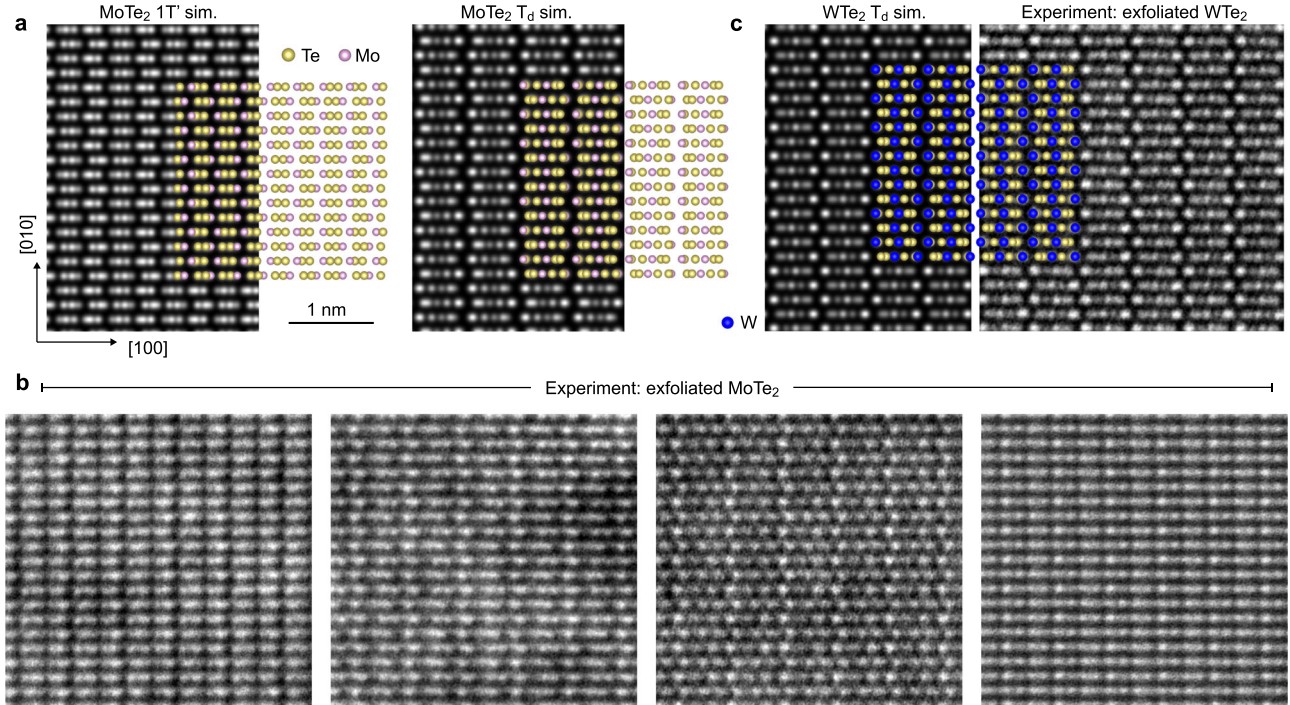

**Fig. 2 | Plan-view atomic-resolution imaging of MoTe₂ and WTe₂. a** Simulated HAADF-STEM images of 1T′ and $T_d$ MoTe₂ in plan-view (*ab*-plane), with overlaid atomic schematics. **b** Several experimental HAADF-STEM images of exfoliated MoTe₂, none of which match the simulated $T_d$ or 1T′ images. The left three panels are all from the same flake, while the rightmost panel is from a separate flake. **c** Simulated and experimental STEM data for exfoliated WTe₂. The experimental data matches the $T_d$ simulation. The scale bar in (**a**) applies to (**b**) and (**c**) as well.

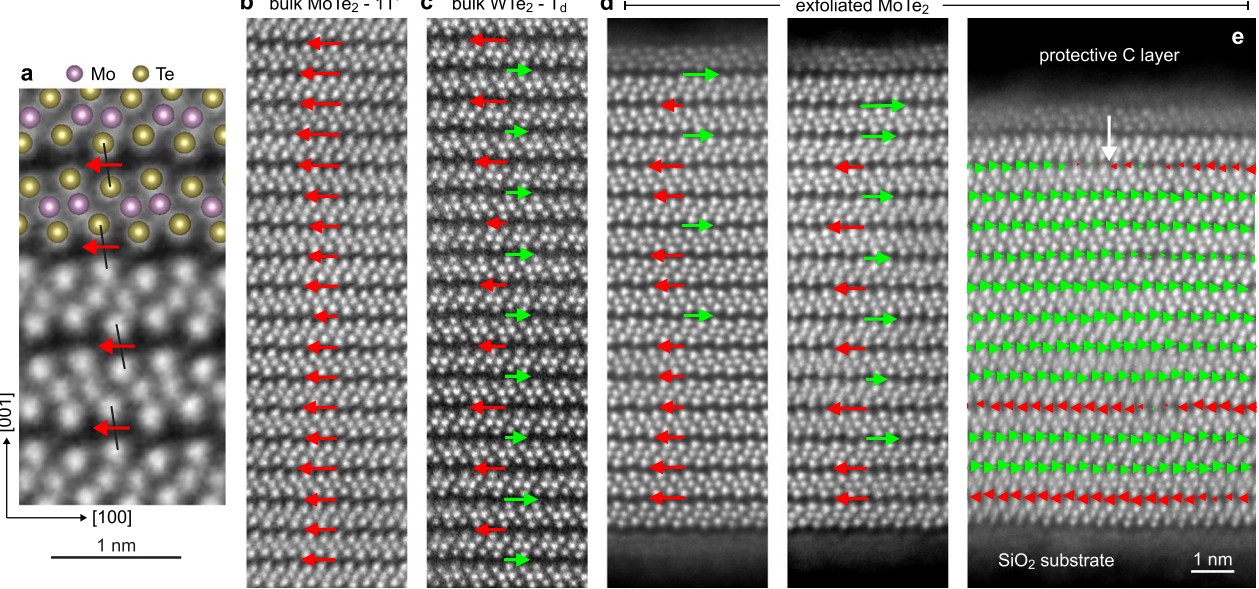

**Fig. 3 | Cross-section atomic-resolution imaging of MoTe₂ and WTe₂.**
**a** Magnified HAADF-STEM image of bulk MoTe₂ with an overlaid atomic structure schematic. The black lines show the bridging Te-Te pairs and the red arrows represent the inter-layer shift. **b, c** HAADF-STEM images of bulk MoTe₂ and WTe₂, respectively, with overlaid arrows representing the average value of **λ**. **d** HAADF-STEM images of a MoTe₂ flake exfoliated onto amorphous SiO₂ with a protective carbon overlayer. The arrow magnitudes in **b–d** are 15 times the calculated shift. **e** HAADF-STEM image of the same MoTe₂ flake, highlighting a stacking soliton (marked with the white arrow). **λ** is represented for each individual Te-Te pair. The size of the arrows correlates with the magnitude of the shift, with a nonlinear scaling to emphasize the stacking soliton; the shift magnitude is suppressed in this region. The scale bar in (**e**) applies to (**b–d**) as well.

direction (↓↓↓↓ or ↑↑↑↑), while for the $T_d$ phase, the shift direction alternates (↑↓↑↓). Fig. 3a shows a HAADF-STEM image of bulk MoTe₂, prepared in the cross-sectional geometry using a focused ion beam (FIB). The inter-layer shift (referred to as **λ** in Fig. 1a) is clearly visible. To quantify the inter-layer shift, we fit all Te columns with a 2D Gaussian and directly calculate **λ** for each Te-Te pair (Supplementary Note 1, Supplementary Fig. 2). We then average **λ** laterally across the image width for each layer. Fig. 3b, c demonstrates this method on bulk crystals of MoTe₂ and WTe₂, respectively. The bulk MoTe₂ structure follows the expected room temperature 1T′ stacking

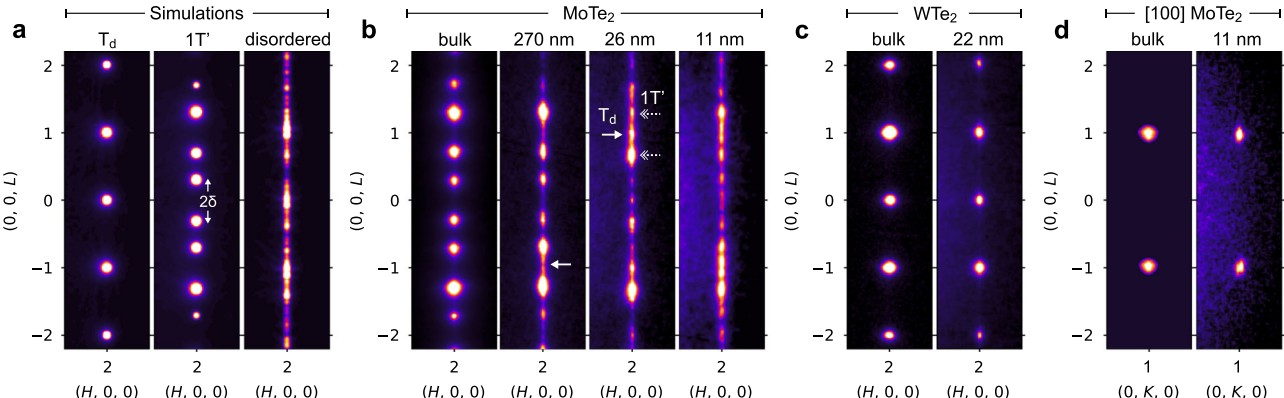

**Fig. 4 | Electron diffraction of MoTe₂ and WTe₂. a** Electron diffraction simulations of MoTe₂ with 1T', T$_d$, and random stacking. Note that all diffraction data is indexed using the orthorhombic T$_d$ unit cell. **b** Experimental TEM selected-area diffraction of MoTe₂ as a function of flake thickness. For the 270 nm flake, the arrow indicates diffuse scattering along L. For the 26 nm flake, the arrows highlight both 1T' and T$_d$ diffraction spots. **c** Experimental data for bulk and thin-flake WTe₂. **d** Experimental data for bulk and thin-flake MoTe₂, viewed down the orthogonal axis. Enlarged 2D electron diffraction datasets are shown in Supplementary Fig. 5.

sequence (↓↓↓↓), and bulk WTe₂ follows the expected T$_d$ stacking sequence (↑↓↑↓). We note that both bulk crystals show some degree of stacking disorder, e.g., twin boundaries (Supplementary Fig. 3).

We next examine STEM data from a ~9 nm thick MoTe₂ flake exfoliated onto a SiO₂/Si substrate. Fig. 3d, e shows different regions from the same flake, with the regions separated laterally by 100 s of nm. In stark contrast to bulk MoTe₂ which shows ordered 1T' stacking, the thin flake of MoTe₂ displays an array of alternative stacking arrangements. There is no strong preference for either 1T' or T$_d$ stacking; the stacking lacks long-range order and appears highly disordered. Changes in layer stacking are accommodated at stacking solitons[32], where the inter-layer shift changes direction. An example soliton is highlighted in Fig. 3e, where arrows representing λ for each individual Te-Te pair are shown. The observed disordered layer stacking explains the inter-layer Raman mode observed in thin MoTe₂ flakes at room temperature (Fig. 1c and refs. 17,19) since certain local stacking sequences break inversion symmetry. Additionally, the mixed and disordered stacking observed in Fig. 3 explains the *ab*-plane STEM data shown in Fig. 2. Specifically, *ab*-plane STEM simulations of mixed stacking arrangements are consistent with the unexpected image contrast observed experimentally by plan-view STEM (Supplementary Fig. 4 and Supplementary Note 2).

To characterize the layer stacking on a more global scale, rather than small regions examined by STEM, we performed electron diffraction measurements using a ~3 micron selected-area aperture. We focus on scattering along (2, 0, L), which allows easy differentiation between the various stacking geometries. Specifically, simulations show that T$_d$ stacking yields diffraction spots at $L = N$ (N is any integer) while 1T' gives spots at $L = N \pm \delta$. The offset $\delta$ is due to the 1T' monoclinic angle of β ~ 93.9°, and the doubling is due to the two twin variants; ↓↓↓↓ and ↑↑↑↑ stacking domains give $L = N + \delta$ and $L = N - \delta$ (Fig. 4a). To capture alternative stacking arrangements which lack long-range order, we constructed a random stacking model by fixing the magnitude of λ at 45 pm, and randomly selecting the shift direction for each new layer (Supplementary Note 3). The random stacking model results in diffuse scattering along $L$[33].

Experimentally, diffraction from bulk MoTe₂ matches the 1T' simulation, as expected (Fig. 4b). For a 270 nm thick MoTe₂ flake, we observed 1T' diffraction spots; however, diffuse scattering along L is also present (see arrow), indicating a measurable degree of stacking disorder. For flakes 26 and 11 nm thick, spots corresponding to both 1T' and T$_d$ are present (see arrows), and the diffuse scattering along L is further enhanced. These diffraction measurements show that stacking disorder in MoTe₂ flakes is a global effect. In contrast to MoTe₂, both bulk and exfoliated WTe₂ show ordered T$_d$ stacking with no apparent

thickness effect (Fig. 4c). This is consistent with our HAADF-STEM imaging of WTe₂ flakes (Fig. 2c).

As an alternative interpretation of the MoTe₂ diffraction data in Fig. 4, the spot streaking could be an intrinsic size effect unrelated to stacking disorder. To test this hypothesis, we measured the same 11 nm thick MoTe₂ flake from Fig. 4b along the orthogonal axis (in the *bc*-plane). As shown in Fig. 4d, there is minimal streaking of (0, 1, L) spots, indicating that intrinsic size effects are negligible and that streaking of the (2, 0, L) spots is related to stacking. Note that when viewed in the *bc*-plane, the bulk 1T' and T$_d$ stacking sequences are indistinguishable, and the bridging Te-Te pairs have no component along the *b*-axis (λ$_b$ = 0 Å). The diffraction data in Fig. 4d is consistent with λ$_b$ = 0 Å, and this is confirmed by HAADF-STEM imaging (Supplementary Fig. 2). Hence, the disordered stacking in thin MoTe₂ flakes is restricted to shifts along the *a*-axis, while *b*-axis shifts are absent, similar to the bulk 1T' and T$_d$ phases.

## Cryogenic electron diffraction

Having established mixed and disordered stacking in thin MoTe₂ flakes at room temperature (but not in thin WTe₂ flakes), we next consider whether well-ordered T$_d$ stacking can be stabilized in MoTe₂ at sufficiently low temperature. We find that FIB sample preparation restricts layer sliding in bulk MoTe₂ (Supplementary Fig. 6), thus our approach of cross-sectional (S)TEM analysis cannot be used to study temperature effects. Instead, flakes must be studied in plan-view[34]. Unfortunately, there is no direct method to determine the layer stacking in this geometry: analysis of plan-view HAADF-STEM imaging is complicated owing to the stacking disorder (Supplementary Fig. 4), and the layer stacking does not influence the (H, K, 0) diffraction pattern symmetry. However, the layer stacking does influence the (H, K, 0) diffraction spot intensities (Supplementary Fig. 7), and we use this relation to infer the layer stacking.

Figure 5a shows *ab*-plane diffraction data for WTe₂ at room temperature, as well as MoTe₂ at both room temperature and -17 K. From electron energy loss spectroscopy analysis, the WTe₂ flake is ~24 nm thick, and the MoTe₂ flake is ~37 nm thick (Supplementary Fig. 8). We first analyze the data qualitatively. Starting with WTe₂, we observe a clear first-order Laue zone, which we highlight in Fig. 5b by plotting the intensity of each diffraction spot as a function of momentum transfer, Q. The Laue zone is indicative of out-of-plane order with a real-space periodicity of 14 Å, in good agreement with the WTe₂ T$_d$ c-lattice parameter. In contrast, for the MoTe₂ flake measured at room temperature, there is no Laue zone, indicating a lack of out-of-plane order. Multi-slice electron diffraction simulations of disordered stacking show a broad suppression of the Laue zone, consistent with the

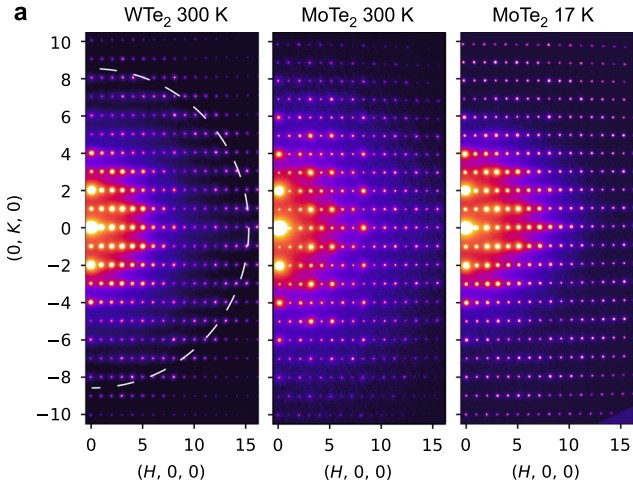

**b**

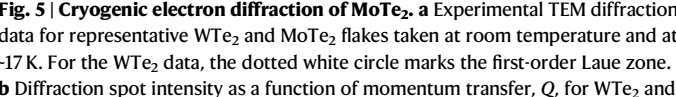

**c** 300 K, Rnd ⟶ 17 K, $T_d$-Rnd

**Fig. 5 | Cryogenic electron diffraction of MoTe₂. a** Experimental TEM diffraction data for representative WTe₂ and MoTe₂ flakes taken at room temperature and at -17 K. For the WTe₂ data, the dotted white circle marks the first-order Laue zone. **b** Diffraction spot intensity as a function of momentum transfer, $Q$, for WTe₂ and MoTe₂. The vertical dashed line marks the first-order Laue Zone for both 1T′ and $T_d$ phases. **c** Schematic of a random stacking sequence (Rnd) versus our $T_d$-Rnd model, which lacks long-range order but possesses short-range $T_d$ order. The brackets highlight local $T_d$ domains.

experimental data (Supplementary Fig. 7). Upon cooling the MoTe₂ flake using liquid He TEM, a weak Laue zone emerges, suggesting the presence of partially ordered layer stacking (Fig. 5a, b).

We next analyze the diffraction data quantitatively. To do so, we extract all the experimental $(H, K, 0)$ diffraction spot intensities with $H \leq 5$ and $K \leq 3$. We then model the spot intensities with multi-slice electron scattering calculations, using the flake thickness, orientation, and bending as fitting parameters. Fits are performed assuming $T_d$, 1T′, and disordered stacking, and the resulting $\chi_v^2$ values are compared (Supplementary Note 4). The results are outlined in Table 1. For WTe₂ measured at room temperature, the analysis strongly favors $T_d$ stacking, in agreement with our STEM data (Fig. 2c). This result demonstrates the validity of our quantitative diffraction approach. For the MoTe₂ flake measured at room temperature, the random stacking model provides an excellent fit, consistent with the cross-sectional (S)TEM data (Figs. 3 and 4). Upon liquid He cooling, the data is best fit with a model that lacks long-range stacking order but favors local $T_d$ stacking, with an average $T_d$ domain thickness of ~6 layers (4 nm). This model, labeled $T_d$-Rnd, is schematically illustrated in Fig. 5c. We conclude that there is an increase in the relative fraction of $T_d$ stacking after cooling to liquid He temperatures, but there is no transition to a fully ordered $T_d$ state. This conclusion is supported by both our qualitative Laue zone analysis and the quantitative multi-slice fitting method.

We also performed in situ annealing on a separate MoTe₂ flake with $ab$-plane diffraction. Starting at room temperature, our quantitative and qualitative analyses both indicate disordered stacking, as expected. Upon warming to 675 K, there were no significant changes in the diffraction spot intensities (Supplementary Fig. 10). Thus, even with high-temperature heating, there is no transition to ordered 1T′ and no measurable changes in the layer stacking.

## Table 1 | $\chi_v^2$ fitting results from our quantitative diffraction approach

| | Temp. (K) | $T_d$ | 1T′ | Random | $T_d$-Rnd |
|---|---|---|---|---|---|
| WTe₂ | 300 | **3.8** | 40 | 50 | - |
| MoTe₂ | 300 | 18 | 15 | **4.3** | - |
| MoTe₂ | 17 | 9.2 | 14 | 4.8 | **2.4** |

The listed temperatures are approximate. Further details are given in Supplementary Note 4 and Supplementary Fig. 9.
For each row, the bold value represents the best fit.

## Discussion

Considering all our experimental data, we summarize our findings below. For flakes ≤10 nm in thickness, the stacking is highly disordered with minimal thermal dependence. For flakes 10s of nm in thickness, the room temperature stacking is also highly disordered. Upon cooling flakes in this thickness range, there is an increase in local $T_d$ order, but the transition is only partial, and the disorder persists down to the lowest measured temperatures. We emphasize that with reduced flake thickness, neither the 1T′ nor $T_d$ structures are stabilized. Rather, with reduced thickness, we observe a transition from ordered to disordered layer stacking and a suppression of thermally induced layer sliding. Lastly, we note that for monolayer and bilayer flakes, there is no distinction between 1T′ and $T_d$ stacking.

### Consideration of extrinsic effects for stacking disorder

We now consider the possible origins of the stacking disorder. The flakes discussed in Figs. 2–5 were exposed to ambient atmosphere prior to (S)TEM analysis, thus oxidation effects may be present. As a control experiment, we studied a flake exfoliated in an Ar glovebox and fully encapsulated with hexagonal boron nitride (h-BN), and this flake similarly showed layer stacking disorder (Supplementary Fig. 11). As a separate control, we imaged the top surface of a bulk crystal exposed to atmosphere (Supplementary Fig. 12), and we observed well-ordered 1T′ stacking up to the topmost layers. Hence, the observed stacking disorder is not due to oxidation. Additionally, the observation of ordered 1T′ stacking at the surface of a bulk crystal (Supplementary Fig. 12) rules out mechanisms related to inherent surface effects, e.g., the breaking of [001] translational symmetry. Interfacial effects from the SiO₂ substrate can also be eliminated based on the h-BN encapsulation experiment and since free-standing MoTe₂ flakes show stacking disorder as well (Figs. 2 and 5).

The starting crystal quality, such as the concentration of vacancies and interstitials, may influence the layer stacking. However, our bulk MoTe₂ crystal displayed well-ordered 1T′ stacking at room temperature (Fig. 3a), as well as a sharp first-order phase transition in electrical resistance (Fig. 1b). Both findings indicate high crystal quality. Moreover, three separate sources of bulk MoTe₂ were tested (including high-quality crystals with typical residual resistivity ratios (RRRs) of ~500[30,33]), and all bulk crystals yielded thin flakes with disordered stacking ("Methods"). Hence, the parent crystal quality cannot explain the disordered stacking found in thin flakes. Electron irradiation effects must also be considered[34]. In our STEM cross-section

experiments, we observed the gradual amorphization of $MoTe_2$ and $WTe_2$ in a layer-by-layer fashion; however, we did not observe any layer sliding under the electron beam. Thus, STEM-induced effects cannot account for the observed disorder. FIB-induced damage can also be ruled out since both bulk and thin-flake samples were prepared with the same FIB procedure, but only the thin flakes displayed disorder.

Scotch tape exfoliation (and the associated mechanical strain) offers a possible explanation for our findings; however, there are several reasons to doubt this mechanism. First, with our experimental method, the normal and shear strains applied to a flake during exfoliation are independent of the flake thickness (Supplementary Note 5). Thus, this hypothesis suggests that all exfoliated flakes, regardless of thickness, should show the same level of disorder. Instead, we find that the disorder is greatly enhanced for thin flakes (Fig. 4). Second, the inter-layer force constant—which would resist any mechanically induced layer sliding—is comparable for $WTe_2$ and $MoTe_2$[19,35]. If mechanical exfoliation were responsible for the disordered stacking in $MoTe_2$, then exfoliated $WTe_2$ should show similar levels of disorder, which is not the case (Figs. 2 and 4). Mechanical strain might also cause disordered shifts along the [010] axis in $MoTe_2$, but this is not observed experimentally (Supplementary Fig. 2). Finally, if the observed disorder were simply due to mechanical strain, then one might expect a high temperature anneal to restore the equilibrium (ordered) stacking arrangement. Instead, we find minimal changes in layer stacking after annealing flakes up to 675 K (Supplementary Fig. 10).

To summarize, the observed layer stacking disorder in thin $MoTe_2$ flakes cannot be attributed to oxidation, interfacial effects, surface effects, crystal quality, electron irradiation, or scotch tape exfoliation.

### Consideration of intrinsic effects for stacking disorder

We next discuss the possibility of intrinsic coupling between the $MoTe_2$ thickness, layer stacking, and free energy. To test this hypothesis, we performed total energy DFT calculations for 1T′ and $T_d$ stacking for bulk crystals, as well as thin films of various thicknesses (Supplementary Table 1). For bulk crystals, our calculations consistently found $T_d$ stacking as the ground state, consistent with experimental data and prior DFT results[9,36]. However, for thin films, the preferred stacking arrangement was dependent upon the chosen vdW correction (we tested Grimme-D3, rev-vdW-DF2 (rev), and rev+U[37–42]). These discrepancies are not wholly unexpected, given the challenging nature of the problem: the layer stacking energy scale is ~1 meV/formula unit, which is approaching the uncertainty of DFT calculations; the layer stacking energetics are determined by vdW interactions, which are difficult to capture with DFT and depend sensitively upon the chosen vdW correction; and for thin flakes, surface and boundary effects present compounding challenges. Thus, we cannot make any definitive claims regarding the stability of 1T′ or $T_d$ stacking for any specific flake thickness. Still, meaningful trends can be extracted from DFT calculations of bulk structures. Specifically, Kim et al. found a larger inter-layer coupling for $MoTe_2$ compared to $WTe_2$[36]. The same study found that upon transitioning from the 1T′ to $T_d$ phase, the electronic bands are strongly altered for $MoTe_2$ but not for $WTe_2$. Taken together, these findings suggest that in thin flakes of $MoTe_2$, the band structure will experience a thickness effect (owing to the strong inter-layer coupling), which will then modulate the layer stacking energetics. In contrast, for $WTe_2$, thickness-effects (and their influence on the stacking energetics) should be reduced. This reasoning is in line with our observations of disordered stacking in thin $MoTe_2$ flakes but ordered $T_d$ stacking in $WTe_2$ flakes.

Alternatively, the disordered stacking in $MoTe_2$ may be an entropic—rather than an energetic—effect. When calculating the total free energy of a material, the phonon energies dictate the vibrational entropy and lower energy phonons yield a lower free energy[43]. In bulk $MoTe_2$, the energy of the inter-layer shear phonon is significantly higher for the $T_d$ phase than the 1T′ phase (1.71 versus 1.55 meV)[44]. Moreover, it was argued by Heikes et al. that the $MoTe_2$ stacking

transition is driven by the differing inter-layer phonon energies and their effect on the vibrational entropy[9]. In the context of our work, if reducing the $MoTe_2$ flake thickness alters the energy of the inter-layer phonon modes, then the relative free energy of 1T′ and $T_d$ stacking would be affected. Indeed, it is well-established that the inter-layer phonon modes in $MoTe_2$ are strongly thickness-dependent[19]. Thus, it is possible that our observations of disordered stacking in $MoTe_2$ are driven by a coupling between thickness, phonon energy, and entropy.

### $MoTe_2$ versus $WTe_2$

A major finding of our work is the contrast between ordered stacking in thin $WTe_2$ flakes and disordered stacking in thin $MoTe_2$ flakes. As noted in the prior section, DFT results from Kim et al. suggest that a stronger coupling between thickness, stacking, and energy for $MoTe_2$ compared to $WTe_2$ may be responsible for this difference. There is also experimental evidence that suggests a more complex stacking landscape in $MoTe_2$ versus $WTe_2$. Specifically, neutron diffraction measurements of bulk single-crystal $MoTe_2$ show that upon cooling, the 1T′ to $T_d$ phase transition occurs through an intermediate disordered phase[33] and that upon warming, the transition occurs through a metastable $T_d^*$ phase, which can be described as ↓↓↑↑ stacking[30]. In contrast, for bulk $WTe_2$ single crystals, the 1T′ to $T_d$ phase transition is abrupt upon both heating and cooling, with no intermediate phases or thermal hysteresis[16]. Hence, even in bulk, there is a propensity for disordered and alternative stacking in $MoTe_2$ compared to $WTe_2$, which is mirrored in our thin flake results.

### Strategies to obtain ordered stacking in $MoTe_2$

For most device applications, ordered 1T′ or $T_d$ stacking is desirable, and strategies to obtain fully ordered stacking should be developed. Based on DFT calculations[36], charge doping can stabilize 1T′ stacking (hole doping) or $T_d$ stacking (electron doping). This proposal should be studied experimentally in thin exfoliated flakes. We note that our data provide strong evidence that surface oxidation does not stabilize ordered 1T′ stacking via hole doping, as previously suggested in the literature[20,21]. Alternatively, to stabilize the $T_d$ phase, the application of a strong out-of-plane electric field may promote the ferroelectric phase due to the electrostatic coupling. Lastly, if the disordered stacking is due to mechanical strain during exfoliation, then films grown by chemical vapor deposition (CVD) should be explored. However, CVD growths primarily yield monolayer and bilayer flakes, wherein 1T′ and $T_d$ stacking orders cannot be defined, or trilayer flakes, wherein only ordered 1T′ (↓↓ or ↑↑) or $T_d$ (↓↑ or ↑↓) stacking arrangements are possible[45–51]. Alternative stacking arrangements require thicker specimens.

In conclusion, we studied the effect of thickness on the layer stacking of exfoliated $MoTe_2$ flakes through atomic-resolution HAADF-STEM imaging, in situ cryogenic TEM, Raman spectroscopy, electronic resistance measurements, and DFT calculations. We found that thin exfoliated flakes of $MoTe_2$ are not well-ordered 1T′ or $T_d$ but rather possess disordered layer stacking. Our results raise important questions regarding the electronic structure, topology, and charge transport mechanisms in exfoliated $MoTe_2$ flakes, as well as how thickness may influence layer stacking in other 2D materials which exhibit stacking-dependent functionality, e.g., magnetic 2D materials. This work also highlights the importance of atomic-scale analysis in determining the structure of 2D materials.

## Methods
### Materials and sample preparation

We tested three bulk $MoTe_2$ crystal sources: commercially purchased flux-grown $MoTe_2$ from 2D Semiconductors, commercially purchased CVT grown $MoTe_2$ from HQ Graphene, and high-quality (typical RRR ~ 500) flux-grown $MoTe_2$ as previously reported in refs. 30,33. Crystals from 2D Semiconductors were used for the experiments and

data presented in the main text. Supplementary Fig. 11 shows STEM data from the $MoTe_2$ crystals described in refs. 30,33, and Supplementary Fig. 13 compares electron diffraction from all three crystal sources. The studied $WTe_2$ was obtained commercially from 2D Semiconductors. Flakes were prepared via conventional scotch tape exfoliation.

## Electronic transport measurements

Flakes were first exfoliated onto $SiO_2$/Si substrates. Electron-beam lithography and thermal evaporation were then used to fabricate 4-probe devices, using 10 nm Cr as the adhesion layer followed by 100 nm of Au. Electronic resistance versus temperature data were collected using a physical property measurement system, using heating and cooling rates of 2 K/min.

## Raman spectroscopy

Flakes were initially exfoliated onto $SiO_2$/Si substrates and then transferred to sapphire substrates using a PPC stamp. The sapphire substrate offers lower background scattering in the low-frequency regime. Measurements were made using a custom-built Raman microspectrometer system[52]. A frequency-stabilized 785-nm laser (Toptica iBEAM-SMART-785-S-WS) was used as the excitation source, which was focused onto the sample using a long-working distance objective (Mitutoyo, NIR, 10×, NA = 0.26). The power of the laser was adjusted to be below 5 mW by a neutral density filter. The Raman signal was collected by the same objective and guided to a spectrograph (Horiba iHR550) and onto a CCD camera (Horiba Syncerity). A set of five narrow-linewidth, reflective volume Bragg grating notch filters (OptiGrate) was used to block the laser to enable measurements of Raman signals down to about $5 \, cm^{-1}$. The Raman signal was spatially filtered by a pair of 75-mm focal length achromatic lens and a 50 μm pinhole before being sent into the spectrograph. The sample was mounted into an optical cryostat (Janis VPF-100), and the pressure of the cryostat was maintained to be below $1 \times 10^{-4}$ Torr during the measurement.

## Scanning transmission electron microscopy

STEM measurements were performed on an aberration-corrected Thermo Fisher Scientific (TFS) Spectra 300 X-FEG, as well as an aberration-corrected TFS Titan Themis 300 X-FEG. We used accelerating voltages ranging from 120 to 300 kV and probe currents of 50–200 pA. Cross-sectional specimens were prepared using a standard Ga focused ion beam (FIB) lift-out procedure on a TFS Helios G4 X FIB, with final thinning performed at 5 kV. Plan-view specimens were prepared using a PDMS stamp to transfer flakes from scotch tape to holey SiN TEM grids.

## STEM-HAADF simulations

The HAADF-STEM image simulations (Fig. 2 and Supplementary Fig. 4) were carried out using the multi-slice method implemented in the autoSTEM module of Computem[53]. For each simulation, the probe was defined with a defocus of 1.7 nm, a spherical aberration coefficient of 1.5 μm, and a convergence semi-angle of 30 mrad at 300 keV to match the experimental conditions. Transmission and probe wave functions of $2048 \times 2048 \, pixel^2$ were used, resulting in a maximum scattering angle of 401 mrad. The inner and outer collection angles of 80 and 220 mrad, respectively, were used. Phonon effects were not included in the simulations. The final images were blurred by a Gaussian kernel with a width of ~0.7 Å to simulate the effect of finite source size.

## Electron diffraction

For $WTe_2$ measurements and the in situ annealing of $MoTe_2$, electron diffraction was performed on a TFS Osiris TEM operated at 200 kV. The annealing experiment was performed using the Protochips Aduro 300DT System. Exfoliated flakes were transferred to the Protochips e-chip using a PPC stamp. For the liquid He

measurements of $MoTe_2$, data were collected on a JEOL ARM operated at 200 kV using a Gatan liquid-helium cooling holder (HCTDT 3010). The sample temperature is measured using a diode ~6 inches away from the sample. For TEM sample preparation, flakes were initially exfoliated onto a $SiO_2$/Si substrate. Then, a lacey carbon grid was placed on top of the $SiO_2$, and a drop of 10% HF solution was applied to etch the $SiO_2$ and release the $MoTe_2$ onto the lacey carbon. The grid was then flushed with DI water.

## Density functional theory

The DFT calculations were performed using the generalized gradient approximation (GGA) exchange-correlation functional of Perdew, Burke, and Ernzerhof (PBE) as implemented in the Vienna Ab initio Simulation Package (VASP). The starting atomic structures for 1T' and $T_d$ $MoTe_2$ were taken from ref. 11, and the structures were fully relaxed. For the Rev+U simulations, we used U = 5 eV for Mo and U = 2 eV for W.

## Data availability

Data are available upon request. The STEM data have been deposited online, https://doi.org/10.34863/bm64-9w84.

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

## Acknowledgements

J.L.H. and J.J.C. were funded through the Gordon and Betty Moore Foundation's EPiQS Initiative, grant GBMF9062.01. The work at the University of Virginia has been supported by the Department of Energy, Grant number DE-FG02-01ER4592. S.L. and P.G. acknowledge the support from the Air Force Office of Scientific Research (Grant No. FA9550-22-1-0209). The work at BNL is supported by the Office of Basic Energy Sciences, Materials Sciences and Engineering Division, U.S. DOE under Contract No. DE-SC0012704. Device fabrication was performed in part at the Cornell NanoScale Facility, a member of the National Nanotechnology Coordinated Infrastructure (NNCI), which is supported by the National Science Foundation (Grant NNCI-2025233). This work made use of the electron microscopy facility of the Platform for the Accelerated Realization, Analysis, and Discovery of Interface Materials (PARADIM), which is supported by the National Science Foundation under Cooperative Agreement No. DMR-2039380, and the Cornell Center for Materials Research Shared Facilities which are supported through the NSF MRSEC program (DMR-1719875). The FEI Titan Themis 300 was acquired through NSF-MRI-1429155, with additional support from Cornell University, the Weill Institute and the Kavli Institute at Cornell. L.B., E.B. and L.F.K. acknowledge support by PARADIM and the Packard Foundation.

## Author contributions

(S)TEM imaging, diffraction experiments, and STEM image simulations were performed by J.L.H., L.B., M.G.H., E.B., and D.J.H. with supervision from Yi.Z., L.F.K., and J.J.C. DFT calculations were performed by Ya.Z., with supervision from E.J.R and F.J. Raman spectroscopy measurements were performed by S.L. and P.G. J.A.S., Y.T., and D.L. provided samples. J.L.H. prepared the electronic devices and performed resistance versus temperature measurements. J.L.H. wrote the manuscript with feedback and contributions from all authors.

## Competing interests

The authors declare no competing interests.
