## [Peer Review File · Nature Communications]

Emergent Layer Stacking Arrangements in c-axis Confined MoTe₂REVIEWER COMMENTS

Reviewer #1 (Remarks to the Author):

The authors studied the layer stacking arrangements of MoTe₂ thin flake and found that highly disordered stacking with nanoscale 1T' and T_d domains, as well as alternative stacking arrangements, which is totally different from its bulk counterpart. Combining atomic-resolution HAADF-STEM imaging, in situ cryogenic TEM, Raman spectroscopy, electronic resistance measurements, and DFT calculations, they plotted the temperature-thickness stacking phase diagram, from which we can find that exfoliated thin flakes of MoTe₂ are not well-ordered 1T' or T_d, but rather possess disordered layer stacking. Furthermore, the authors discussed the origin of disordered stacking by DFT calculations, which cannot well explain the disordered stacking. Finally, they proposed a synergistic effect of thickness, phonon energy and entropy. Overall, the experimental results are novel and interesting, which are crucial for understanding the exotic quantum properties in MoTe₂ flakes. Therefore, it deserves to be published in Nature Communications. Before acceptance, several issues should be addressed:

1. The authors claimed that their samples are of high crystal quality, but in their Raman data, compared to refs. [19] and [21], the characteristic peaks show obvious broadening and weakening, which is regarded as sample degradation or oxidation, though the authors claimed the observed stacking disorder is not due to oxidation.
2. I concern that the authors' observation of stacking disorder depends on the specific synthesized method. Though three different sample sources were used in the authors' experiments, all the measured samples are cleaved from their bulk counterparts. A prior STEM study performed on the CVD prepared MoTe₂ [Nat. Commun. 10, 2044 (2019)] showed that the ultra-thin (<10 nm) MoTe₂ samples crystallize in T_d-phase. To improve the reliability and reproducibility, I suggest the authors to carry out control experiments on more samples, including high-quality crystals and non-exfoliated CVD samples.
3. The authors argued that the interfacial effects from the SiO₂ substrate can be eliminated by observing the same stacking disorder in free-standing MoTe₂ flakes, which is not convincing. It is well known that the interfacial effects from substrate can not be negligible for ultrathin materials. As shown in ref. [21], golden substrate strongly interacts with MoTe₂ flakes and leads to a different phase of MoTe₂ thin film. Therefore, the authors' observation of stacking disorder with substrate can originate from interfacial effects. While the stacking disorder observed in free-standing samples may be introduced in mechanical exfoliation or transfer process, not due to strain but some other uncontrollable factors.
4. Previous reports show that it is easily introducing Te vacancy when exfoliating, transferring, heating samples or performing STEM and electron diffraction measurements. But in the manuscript, Te vacancy is almost indiscernible. How did the authors avoid introducing the vacancy?
5. STEM is a local method to characterize sample quality. Did the authors perform global characterizations such as EDX element mapping to show the uniformity of samples, especially in thin flakes? Does sample uniformity affect the stacking order?
6. The disordered or mixed stacking order in thin flakes may be a general phenomenon, due to the small energy difference between 1T' and T_d phase. If we improve sample quality and uniformity, can we get a pure 1T' or T_d phase in thin flakes?
7. The authors seem to omit an important reference previously published in Nat. Commun. 10, 2044 (2019), where similar electronic transport and Raman data in Fig. 1 of the manuscript has been reported in MoTe₂ flake, together with a kind of asymmetric spin-orbit coupling.

Minor comments:

1. There are some typos in the manuscript, for example, in the Electron diffraction, "Exfoliated lakes", "data was collected", etc.

Reviewer #2 (Remarks to the Author):

In this manuscript, the authors mainly used STEM imaging, including both in-plane and cross section imaging, and electron diffraction to study the interlayer stacking of MoTe₂ and WTe₂. Electrical transport, Raman spectroscopy, image simulation and DFT calculations were also used to characterize the crystalline structure and

interlayer stacking. The authors found that the crystalline structure of WTe₂ is well ordered even in its thin layers. However, the interlayer stacking of MoTe₂ thin flakes is complicated, and it is hard to find long range order in either 1T' or Td phase. The authors ruled out the impact from substrates, degradation, crystal quality, etc. The authors used DFT to help understand the stacking of MoTe₂ and WTe₂ from energy point of view.

The experimental results (TEM imaging and electron diffraction) are of high quality, which I appreciate. However, I would recommend submit this manuscript to a more specialized journal for the following reasons:

1. It seems like the findings (1T' or Td in thin flakes of MoTe₂, and stacking disorders not observed in the bulk) have already been expected from the literature, such as references 17-22 cited in the manuscript. This affects the impact of this manuscript.
2. The highly disordered stacking in MoTe₂ thin layers has not been seen in most other 2D materials.
3. From the results presented in this manuscript, it seems that applications of MoTe₂ thin layers in nano devices would be rather difficult because it is hard to control the stacking order and achieve only one single phase in MoTe₂ flakes. The impact of the findings in this work could be broadened if some control methods for achieving single stacking order are proposed/studied.

Other than that, I think the manuscript is well written, and it deserves publication in a more specialized journal.

Reviewer #3 (Remarks to the Author):

In this report, J. Hart et al. explore the interlayer disorder in few-layer and bulk crystals of 1T'/Td-MoTe₂ and compare this disorder to WTe₂. By using cross-sectional TEM they find that the interlayer disorder in WTe₂ is essentially non-existent, while the interlayer disorder in MoTe₂ increases as the material is thinned down. This work elucidates the origins of changes to the structural transition and concomitant resistance observed by others in the field. The results are timely, as work on few-layer MoTe₂ has recently picked up and become a broader interest for the community. I recommend that this work be published with minor revisions. My concerns and questions are outlined below:

1. The authors state χ^2 values for fittings to different types of disorder and stacking for their TEM observations, however they do not state the reduced χ^2 values. These values are important for knowing whether the fittings themselves are valid. Could the authors please state what they are.
2. The authors show order/disorder via cross-sectional STEM in few-layer MoTe₂, but how does this compare to the bulk crystal for low temperature? Does it have the same structural disorder problems to the Td structure?
3. Have the authors checked CVD grown samples of 1T'-MoTe₂, where no exfoliation was taken? This would be a better way of ruling out any effects from the exfoliation process.
4. What happens to the stacking order at low temperature when the exfoliated flakes are thinner than the average Td domain size? Why was this study limited to looking at flakes of 7 nm or greater?
5. Figure 6 is in a variety of ways, misleading and not necessary for the story of the paper. It is incredibly speculative and should not be in the main text. The figure suggests things like gradual boundaries and transitions back to an ordered T' state at higher temperatures for exfoliated flakes and is directly at odds with what is stated for the experimental observations in the paper (i.e., that no return from a disordered state at RT to an ordered 1T' structure is observed, regardless of increased temperature). The same can be said for the Td part of this phase diagram. The diagram also indicates many additional points for which the data is not given in either the main text or the supplementary. In some cases, the diagram points to even thinner samples than anything shown in the figures (down to 6 nm for R vs T for instance). It is also not clear how the authors can say that the samples are in the mixed state versus the fully disordered state solely from the R vs T curves (as indicated on the figure).
6. Similar problems exist for the authors' table 2. Many of the values are inconsistent with experimental observations (particularly for WTe₂). The authors admit this in the main manuscript. While it's important to check these calculations and see if anything can be gleaned from them, I'm not convinced that they are important for proving the main hypothesis of the manuscript and find them distracting from the main message instead. The discussion around the values calculated in table 2 could be much more concise and the table itself included in the supplementary instead.

7. On line 312, the authors state that Fig 1b is indicative of high quality MoTe₂ crystals. However, their residual resistivity ratios from Fig. 1b for the bulk crystal is just 10. This indicates an incredible amount of defects in the material, likely in-plane point defects. The authors argue that this disorder does not impact their results, stating that they have explored crystals from a variety of providers and growth conditions. Again, none of these results are shown in the main manuscript or supplementary and it would be great to know what kind of comparisons the authors made between differing quality material.

8. Finally, I would like to note that the graphite encapsulation utilized by the authors is insufficient at preventing oxidation of few-layer, air-sensitive TMDs. H₂O/O₂ can still easily diffuse through the SiO₂/MoTe₂ interface and even under the graphite/MoTe₂ interface unless the TMD is fully encapsulated on all sides.

We thank the reviewers for their insightful questions and suggestions. Below are our point-by-point responses to their feedback. For clarity, reviewers' comments are marked in black and our responses in blue. Revisions in the manuscript are marked in red in the main text and Supplementary Information for convenience, but not included in the response letter for brevity.

Reviewer #1 (Remarks to the Author):

The authors studied the layer stacking arrangements of MoTe₂ thin flake and found that highly disordered stacking with nanoscale 1T' and T_d domains, as well as alternative stacking arrangements, which is totally different from its bulk counterpart. Combining atomic-resolution HAADF-STEM imaging, in situ cryogenic TEM, Raman spectroscopy, electronic resistance measurements, and DFT calculations, they plotted the temperature-thickness stacking phase diagram, from which we can find that exfoliated thin flakes of MoTe₂ are not well-ordered 1T' or T_d, but rather possess disordered layer stacking. Furthermore, the authors discussed the origin of disordered stacking by DFT calculations, which cannot well explain the disordered stacking. Finally, they proposed a synergistic effect of thickness, phonon energy and entropy. Overall, the experimental results are novel and interesting, which are crucial for understanding the exotic quantum properties in MoTe₂ flakes. Therefore, it deserves to be published in Nature Communications. Before acceptance, several issues should be addressed:

We thank the reviewer for the overall positive assessment of our work and for insightful questions. Below are our responses to the reviewer's comments.

1. The authors claimed that their samples are of high crystal quality, but in their Raman data, compared to refs. [19] and [21], the characteristic peaks show obvious broadening and weakening, which is regarded as sample degradation or oxidation, though the authors claimed the observed stacking disorder is not due to oxidation.

RESPONSE 1.1: We do not believe that our Raman data indicates oxidized MoTe₂ based on two reasons. First, the broadening of our Raman data is consistent with that of h-BN encapsulated MoTe₂ (from ref. 17), and second, layer stacking disorder provides a natural explanation for peak broadening *without* any oxidation or degradation. Lastly, we have carried out further experiments and observed stacking disorder in hBN-encapsulated MoTe₂ flakes, which is addressed in our response to Reviewer 1's question 3.

In Response Fig. 1a we have copied Raman data from ref. 17, which investigated h-BN encapsulated MoTe₂, *i.e.*, MoTe₂ completely free from oxidation. Here, we focus on the data from ref. 17, rather than from refs. 19 and 21 because the MoTe₂ in both refs. 19 and 21 were exposed to oxygen. We focus on the inter-layer shear mode (peak A) which is most sensitive to the layer stacking order. The peaks from ref. 17 have a full-width at half maximum (FWHM) in the range of 3 – 5 cm⁻¹. In Response Fig. 1b, we plot our Raman data. The circular markers show the raw experimental data, and the lines show fits, based on a decaying background function and a Gaussian peak. For the 7 nm thick flake, the fitted FWHM of the inter-layer shear mode is 3.8 cm⁻¹.

¹. Hence, the width of our Raman data is consistent with that of encapsulated MoTe₂ in ref. 17, free from any oxidation or environmental degradation. The intensity of our interlayer shear mode is lower than that of ref. 17, but the peak intensity is sensitive to many experimental parameters and cannot be quantitatively compared.

Response Figure 1. **a** Copied data from ref. 17 of the main text. This data corresponds to a 4.5 nm thick MoTe₂ flake encapsulated in h-BN, and thus free from any oxidation. The width of the interlayer shear mode is ~ 4 cm⁻¹. **b** Our Raman data of MoTe₂ flakes. The circular markers are the experimental data points, and the lines are fits, consisting of a decaying background function and a Gaussian peak. For the 7 nm thick flake, the fitted FWHM of the shear mode is 3.8 cm⁻¹. Hence, the broadening of our data is fully consistent with h-BN encapsulated flakes.

Second, we note that peak broadening is expected from layer stacking disorder. The energy of the inter-layer shear mode is sensitive to the layer stacking. Specifically, neutron scattering measurements have shown that for the bulk 1T', T_d*, and T_d phases, the inter-layer shear mode is 12.5, 11.9, and 13.8 cm⁻¹, respectively (see ref. 44 of main text). Therefore, alternative stacking sequences distinct from the 1T', T_d*, and T_d phases would produce different shear mode energies as well. In our Raman experiment, we use a \sim micron scale spot size which likely averages over several distinct layer stacking arrangements, all with slightly different shear mode energies. The width of the measured peak will then be broadened according to the differing shear mode energies.

To conclude, our Raman data is consistent with that of the h-BN encapsulated flakes in ref. 17. Additionally, for both our data and the data in ref. 17, the broadened Raman features can be explained in terms of layer stacking disorder. Finally, further experiments on hBN-encapsulated MoTe₂ flakes show layer stacking disorder, which is added as Supplementary Figure 11.

2. I concern that the authors' observation of stacking disorder depends on the specific synthesized method. Though three different sample sources were used in the authors' experiments, all the measured samples are cleaved from their bulk counterparts. A prior STEM study performed on the CVD prepared MoTe₂ [Nat. Commun. 10, 2044 (2019)] showed that the ultra-thin (<10 nm) MoTe₂ samples crystallize in T_d-phase. To improve

the reliability and reproducibility, I suggest the authors to carry out control experiments on more samples, including high-quality crystals and non-exfoliated CVD samples.

RESPONSE 1.2: We appreciate and agree with the reviewer’s point that thin CVD grown crystals may yield layer stacking arrangements distinct from exfoliated flakes. This is an interesting question that we have also considered. Unfortunately, we have concluded that it is not feasible to test this hypothesis. The challenge is that high-quality CVD growth primarily yields monolayers or bilayers, occasionally 3- or 4-layer flakes, but rarely films > 5 nm. Conversely, to observe layer stacking disorder, it is necessary to study flakes at least several nm thick. For instance, a monolayer of MoTe₂ has no layer stacking. A bilayer is characterized by a single layer stacking shift, either ↑ or ↓, but cannot be classified as 1T′ or T_d. A 3-layer flake can be 1T′ (↓↓ or ↑↑) or T_d (↓↑ or ↑↓), but alternative stacking arrangements are not possible. A 4-layer flake has 2³ = 8 possible stacking arrangements, 4 of which are ordered 1T′ and T_d, and the other 4 arrangements are mixed / disordered. For thicker flakes, the number of ordered stacking arrangements remains fixed at 4 (↓↓↓..., ↑↑↑..., ↓↑↓..., ↑↓↑...), but the number of mixed / disordered stacking arrangements grows exponentially. Hence, if disordered stacking is present, it is best studied in flakes >> 4 layers thick. While CVD growths of the tellurides have been steadily increasing, it is still the case that octahedral MoTe₂ with large grain sizes and layer control with thicknesses >> 4 layers is virtually non-existent.

Response Table 1 summarizes several recent CVD growth studies of MoTe₂. The CVD grown films are predominantly mono and bilayer, with no flakes well-suited for the investigation of layer stacking disorder. We have added these references and associated discussion to the main text, see lines 378 – 382.

Response Table 1. Summary of MoTe₂ CVD growth studies. We note that these studies are limited to thicknesses of just several layers

Paper Title	Journal / Year	Flake thickness	Lateral size
Lithography-free high-density MoTe ₂ nanoribbon arrays	Materials Today 2022	~3 layers	50 nm x 100 μm
Growth of bilayer MoTe ₂ single crystals with strong non-linear Hall effect	Nature Communications 2022	1, 2, and 3 layers	10 – 100+ μm
Synthesis of large-scale monolayer 1T′-MoTe ₂ and its stabilization via scalable hBN encapsulation	ACS Nano 2021	Monolayer	10 – 100+ μm
Controlled growth of large-scale uniform 1T′ MoTe ₂ crystals with tunable thickness and their photodetector applications	Nanoscale Horizons 2020	1 – 4 layers	10 – 100 μm
A Simple Method for Synthesis of High-Quality mm-scale 1T′ transition-metal telluride and near-field nanooptical properties	Advanced Materials 2017	Single to few layers	10 – 100 μm
Transport evidence of asymmetric spin-orbit coupling in few-layer superconducting 1T _d -MoTe ₂	Nature Communications 2019	Single to few layers	10 – 100 μm
Large-Area and High-Quality 2D Transition Metal Telluride	Advanced Materials 2016	Single to 10 layers*	10 – 100+ μm

*This study is focused on mono and bilayer MoTe₂. There is a Raman dataset of a 10-layer thick flake (Fig. 2d); however, there were no additional characterizations to verify the thickness.

We now discuss the CVD grown MoTe₂ in *Nat. Commun.* **10**, 2044 (2019). In this paper, the structural characterization of layer stacking is plan-view STEM imaging, which we copy below in Response Fig. 2a. This is not a reliable method of layer stacking determination in MoTe₂. We have performed STEM image simulations which show that alternative layer stacking arrangements – such as twinned 1T' stacking or mixed 1T' and T_d stacking – can yield plan-view STEM images similar to that of ordered T_d stacking. These simulations are presented in Supplementary Figure 4 and also copied below in Response Fig. 2b. Hence, the data in the *Nat. Commun.* manuscript does not unambiguously support the claim of T_d stacking in CVD grown MoTe₂.

Response Figure 2. **a** The STEM data and associated simulations from *Nat. Commun.* **10**, 2044 (2019). **b** Our STEM multi-slice simulations (copied from Supplementary Figure 4) showing that alternative stacking sequences, including twinned 1T' and mixed 1T' and T_d, can produce plan-view STEM images similar to that of ordered T_d stacking.

Lastly, regarding the referee's comment that we should carry out additional control experiments using high-quality crystals, we have performed atomic-resolution imaging of thin flakes exfoliated from bulk crystals with typical residual resistivity ratios (RRRs) of ~500. These crystals are described in refs. 30 and 33 of the main text. The updated data is shown in Supplementary Fig. 11 (which is copied in Response Figure 3). The results clearly show the layer stacking disorder in flakes exfoliated from a high-quality bulk crystal.

3. The authors argued that the interfacial effects from the SiO_2 substrate can be eliminated by observing the same stacking disorder in free-standing MoTe_2 flakes, which is not convincing. It is well known that the interfacial effects from substrate can not be negligible for ultrathin materials. As shown in ref.[21], golden substrate strongly interacts with MoTe_2 flakes and leads to a different phase of MoTe_2 thin film. Therefore, the authors' observation of stacking disorder with substrate can originate from interfacial effects. While the stacking disorder observed in free-standing samples may be introduced in mechanical exfoliation or transfer process, not due to strain but some other uncontrollable factors.

RESPONSE 1.3: We agree with the referee's argument that SiO_2 substrates can have a non-negligible effect on 2D materials. We have thus studied MoTe_2 flakes fully encapsulated with h-BN, prepared in an Ar glovebox. The results are shown in Response Fig. 3 and have also been added to our Supplementary Information (Supplementary Figure 11). The layer stacking disorder is present for the h-BN encapsulated MoTe_2 . We note that h-BN encapsulation is widely regarded as the ideal methodology to protect 2D materials. Hence, we have shown that disordered stacking is present for free-standing flakes, flakes on SiO_2 , flakes encapsulated with h-BN, and flakes capped with graphite. These experiments confirm our prior conclusion that the observed disorder is not an interfacial effect.

Response Figure 3. a Example resistance versus temperature data for a crystal grown with the conditions described in refs. 30 and 33 of the main text. For this crystal, the RRR = 478. A separate

crystal showed $RRR = 797$. **b** Optical image of a MoTe_2 flake with top and bottom hexagonal boron nitride encapsulation layers. Exfoliation and heterostructure assembly were performed in an Ar glovebox with O_2 and H_2O levels < 0.5 ppm, using MoTe_2 crystals as described in refs. 30 and 33 of the main text. **c** HAADF-STEM image of the same heterostructure, demonstrating the presence of layer stacking disorder.

4. Previous reports show that it is easily introducing Te vacancy when exfoliating, transferring, heating samples or performing STEM and electron diffraction measurements. But in the manuscript, Te vacancy is almost indiscernible. How did the authors avoid introducing the vacancy?

RESPONSE 1.4: We are unaware of any reports that exfoliating MoTe_2 in an Ar glovebox at room temperature leads to Te vacancies. We are also unaware of any data suggesting that Te vacancies are introduced during PDMS transfer at room temperature. Thus we do not expect the introduction of Te vacancies from mechanical exfoliation or flake transfer.

There are several studies of electron beam-induced Te vacancy formation in MoTe_2 :

- *ACS Appl. Nano Mater* **2**, 3262–3270 (2019)
- *ACS Nano* **11**, 11005–11014 (2017)
- *J. Phys. Chem. C* **125**, 13601–13609 (2021)

All these studies are focused on thin flakes, ranging from 1 – 3 layers. In contrast, our STEM specimens are much thicker (> 10 nm). This has two important implications. First, for a monolayer of MoTe_2 , HAADF-STEM is sensitive to isolated Te vacancies. Conversely, for a specimen > 10 nm thick, a single Te vacancy is likely indiscernible. Thus, it is possible that the STEM probe forms Te vacancies in our specimens, but they are not detectable given our imaging conditions. Secondly, the mechanism of degradation in MoTe_2 is thought to be beam-induced sample etching. This involves a chemical reaction between the MoTe_2 surface and chemical species present within the STEM column, catalyzed by the electron beam (*J. Phys. Chem. C* **125**, 13601–13609 (2021)). This is a surface mechanism. Thus, for thicker specimens with a lower surface to volume ratio, the rate of Te vacancy formation (per unit volume) will be reduced.

Regarding annealing induced Te vacancy formation, prior work has shown that Te vacancies will form at temperatures of ~ 200 °C or higher (*ACS Nano* **11**, 11005-11014 (2017)). Hence, during our preparation and handling of thin MoTe_2 flakes, we never heated samples above 130 °C. We also performed *in situ* TEM annealing of certain MoTe_2 flakes up to ~ 400 °C (675 K), in order to study possible changes in layer stacking (see Supplementary Figure 10). We observed no obvious evidence of sample degradation during this experiment. However, the sample was only studied with electron diffraction, which is not sensitive to low levels of Te vacancies.

5. STEM is a local method to characterize sample quality. Did the authors perform global characterizations such as EDX element mapping to show the uniformity of samples,

especially in thin flakes? Does sample uniformity affect the stacking order?

RESPONSE 1.5: Response Figure 4 shows global characterization of an exfoliated MoTe₂ flake, demonstrating the uniformity of the flake. Panel a shows a HAADF-STEM image of the flake over a hole in the SiN membrane, panels b and c show STEM-EDX mapping, and panel d shows electron diffraction taken over the same hole. The STEM-EDX and diffraction data clearly show that there are no secondary phases present, nor any significant non-uniformities in the sample stoichiometry. We note that the sensitivity of STEM-EDX is rather poor, and this technique is not able to detect local variation in Te vacancies, or other atomic-scale defects. Overall, we find that all of the observed MoTe₂ flakes are highly uniform, with no detectable secondary phases or changes in stoichiometry. The only non-uniformity in flake structure corresponds to the changes in layer stacking, as shown in Fig. 3 of the main text.

Response Figure 4. Global characterization of an exfoliated MoTe₂ flake in plan-view. **a** HAADF-STEM of the MoTe₂ flake, draped over a hole in a SiN / Au membrane. **b, c** STEM-EDX intensity maps of Mo and Te, respectively. **d** Selected area electron diffraction taken over the hole.

6. The disordered or mixed stacking order in thin flakes may be a general phenomenon, due to the small energy difference between 1T' and T_d phase. If we improve sample quality and uniformity, can we get a pure 1T' or T_d phase in thin flakes?

Response 1.6: We agree with the reviewer that stacking disorder in thin MoTe₂ flakes may be a general phenomenon, in part due to the small energy difference between 1T' and T_d phases. Regarding the quality of the bulk crystals, we have studied high-quality crystals with RRR of several hundred, and such crystals still show disordered stacking for exfoliated flakes. Thus, we find no evidence that improved sample quality or uniformity will yield ordered stacking in exfoliated MoTe₂ flakes. Nevertheless, there may be alternative approaches to remove disordered stacking from thin MoTe₂. We have added a paragraph to the *Discussion* in the main text, which considers alternative methods to obtain ordered stacking in MoTe₂ flakes. Please see lines 371 – 382 of the updated manuscript.

7. The authors seem to omit an important reference previously published in Nat. Commun. 10, 2044 (2019), where similar electronic transport and Raman data in Fig. 1 of the manuscript has been reported in MoTe₂ flake, together with a kind of asymmetric

spin-orbit coupling.

Response 1.7: We agree that this is an important paper, and we have added the reference to our manuscript (ref. 47). We note that the layer stacking characterization in this paper is limited as that was not the focus of this paper. The authors provide a plan-view STEM image to support their claim of T_d stacking, but this image could very well correspond to alternative stacking orders. This is described in detail in our Response 1.2.

Minor comments:

1. There are some typos in the manuscript, for example, in the Electron diffraction, “Exfoliated lakes”, “data was collected”, etc.

Thank you for catching these typos. We have corrected them and carefully read the manuscript to ensure no other errors are present.

Reviewer #2 (Remarks to the Author):

In this manuscript, the authors mainly used STEM imaging, including both in-plane and cross section imaging, and electron diffraction to study the interlayer stacking of MoTe_2 and WTe_2 . Electrical transport, Raman spectroscopy, image simulation and DFT calculations were also used to characterize the crystalline structure and interlayer stacking. The authors found that the crystalline structure of WTe_2 is well ordered even in its thin layers. However, the interlayer stacking of MoTe_2 thin flakes is complicated, and it is hard to find long range order in either $1T'$ or T_d phase. The authors ruled out the impact from substrates, degradation, crystal quality, etc. The authors used DFT to help understand the stacking of MoTe_2 and WTe_2 from energy point of view.

The experimental results (TEM imaging and electron diffraction) are of high quality, which I appreciate. However, I would recommend submit this manuscript to a more specialized journal for the following reasons:

1. It seems like the findings ($1T'$ or T_d in thin flakes of MoTe_2 , and stacking disorders not observed in the bulk) have already been expected from the literature, such as references 17-22 cited in the manuscript. This affects the impact of this manuscript.

RESPONSE 2.1: We disagree with the reviewer that the results we obtained were expected given previous studies. References 17-22 (as well as ref. 47) reflect an active debate in the literature regarding the nature of layer stacking in thin flakes of MoTe_2 . The debate has been framed in the context of *ordered* $1T'$ stacking versus *ordered* T_d stacking, with very little consideration of alternative stacking sequences or disordered stacking. These studies are summarized in Response Table 2. Reference 19 is a partial exception, and the authors discuss Raman spectra which “correspond to neither $1T'$ nor T_d phase” and “suggest that several metastable phases exist with similar total energies.” Still, ref. 19 indicates that thin flakes of MoTe_2 primarily adopt ordered

1T' or T_d stacking, with the alternative metastable phases being outliers. Hence, the literature has assumed ordered stacking in thin flakes of MoTe₂, and no studies have suggested (or even considered) that the dominant stacking motif in thin MoTe₂ is disorder. Reviewer 2's assertion that our findings were *expected* contradicts the literature. We argue that our study is the first study to discuss the ubiquity of layer disorder in MoTe₂, which has broad implications for transport properties and potential device applications.

We also emphasize that all of the studies summarized in Response Table 2 are based on Raman, transport, or other indirect characterization methods. We provided the first study to directly image the layer stacking (dis)order in thin MoTe₂ and thus determine the true structure.

Response Table 2. Summary of the literature debate on layer stacking in thin MoTe₂

Ref. #	Title	Relevant conclusions and claims	Considers disorder?
17	Dimensionality-driven orthorhombic MoTe ₂ at room temperature	h-BN capped MoTe ₂ flakes < 12 nm follow T _d stacking up to 400 K, which is driven by thickness-induced band renormalization	No
18	Enhanced Superconductivity in Monolayer T _d - MoTe ₂	Few-layer MoTe ₂ (h-BN encapsulated) follow T _d stacking	No
19	Structural Phase Transition and Interlayer Coupling in Few-Layer 1T' and T _d MoTe ₂	MoTe ₂ flakes (uncapped) with thicknesses of 1 – 8 layers are found in both the 1T' and T _d phases at room temperature. The stacking phase transition is suppressed in thin flakes	Yes: “Raman spectra of intermediate phases that correspond to neither 1T' nor T _d phase with different interlayer vibration modes were observed, which suggests that several metastable phases exist with similar total energies”
20	Tailoring the phase transition and electron-phonon coupling in 1T' MoTe ₂ by charge doping: A Raman study	Thin (uncapped) MoTe ₂ flakes adopt 1T' stacking due to oxidation (and hole doping). Subsequent charge doping can deterministically induce 1T' or T _d stacking order	No
21	Persistence of Monoclinic Crystal Structure in 3D Second-Order Topological Insulator Candidate 1T' MoTe ₂ Thin Flake Without Structural Phase Transition	Thin (uncapped) MoTe ₂ flakes adopt 1T' stacking due to hole doping, caused by the Au substrate and surface oxidation	No
22	Barkhausen effect in the first order structural phase transition in type-II Weyl semimetal MoTe ₂	For thinner (uncapped) flakes, the stacking transition hysteresis is widened, due to pinning effects. Transition completely suppressed for flakes < 6 nm	No
47	Transport evidence of asymmetric spin-orbit coupling in few-layer superconducting 1T _d -MoTe ₂	Few-layer CVD grown flakes follow T _d stacking	No

2. The highly disordered stacking in MoTe₂ thin layers has not been seen in most other 2D materials.

RESPONSE 2.2: As noted in our Response 2.1, the vast majority of prior MoTe₂ papers assumed well-ordered layer stacking in thin flakes based on indirect characterizations. This does not mean disordered stacking does not exist, but rather it was never considered! We provided the first cross-sectional atomic-resolution study of thin MoTe₂, and in doing so, we directly proved that the layer stacking is highly disordered. The stacking in other 2D materials is currently assumed to be well-ordered; however, there have been exceedingly few studies that investigate the layer stacking at atomic resolution. As we argued in the manuscript, the standard methods used to infer layer stacking – Raman spectroscopy and electron transport – are indirect, and thus not 100% reliable. We have performed Raman spectroscopy and electron transport measurements (Figure 1 in main text), which show consistent results with literature; yet we directly show the layer disorder. Hence, it is possible that disordered stacking is a more general phenomena in exfoliated 2D materials, and not specific to MoTe₂, but this has not been considered by the community.

We have several on-going projects (unpublished) which show disordered stacking in other 2D materials: in thin exfoliated flakes of RuCl₃, we observe disordered stacking, and in this system, the stacking influences the magnetic order; in certain rare-earth tri-tellurides, we observe disordered layer stacking which couples to the symmetry of the charge density wave; and in TaS₂, we find that stacking solitons nucleate the charge density wave transition. The layer stacking disorder in these systems is not completely analogous to the layer disorder found in MoTe₂. Still, we have repeatedly observed that layer stacking disorder is not unique to MoTe₂, and that stacking disorder can strongly influence material properties, *e.g.* magnetism, charge density waves, and metal-to-insulator transitions.

3. From the results presented in this manuscript, it seems that applications of MoTe₂ thin layers in nano devices would be rather difficult because it is hard to control the stacking order and achieve only one single phase in MoTe₂ flakes. The impact of the findings in this work could be broadened if some control methods for achieving single stacking order are proposed/studied.

RESPONSE 2.3: We thank the reviewer for this comment, and we agree the manuscript would be strengthened from a discussion of possible strategies to obtain ordered stacking in MoTe₂. A section has been added to the *Discussion* of the main text, lines 371 – 382. Specifically, we consider charge doping, application of an electric field, and CVD growth as possible avenues to obtain ordered stacking in thin MoTe₂ flakes.

Other than that, I think the manuscript is well written, and it deserves publication in a more specialized journal.

We hope that the referee will reconsider the assessment of our work based on our response. We re-iterate that our study is the first to reveal the presence of layer disorder in exfoliated MoTe₂ and that the layer disorder was not expected nor guessed in the literature prior to our study.

Reviewer #3 (Remarks to the Author):

In this report, J. Hart et al. explore the interlayer disorder in few-layer and bulk crystals of 1T'/Td-MoTe₂ and compare this disorder to WTe₂. By using cross-sectional TEM they find that the interlayer disorder in WTe₂ is essentially non-existent, while the interlayer disorder in MoTe₂ increases as the material is thinned down. This work elucidates the origins of changes to the structural transition and concomitant resistance observed by others in the field. The results are timely, as work on few-layer MoTe₂ has recently picked up and become a broader interest for the community. I recommend that this work be published with minor revisions. My concerns and questions are outlined below:

We thank the referee for the positive assessment of our work. Below is our point-by-point response to the referee's comments.

1. The authors state χ^2 values for fittings to different types of disorder and stacking for their TEM observations, however they do not state the reduced χ^2 values. These values are important for knowing whether the fittings themselves are valid. Could the authors please state what they are.

RESPONSE 3.1: We thank the reviewer for this comment. Our reported χ^2 values were, in fact, reduced χ_v^2 values. Equation 2 in the Supplementary Information was incorrect. The corrected Equation 2 is shown below:

$$\text{Equation 2: } \chi_v^2 = \frac{\sum \left(\frac{I_{exp} - I_{sim}}{\sigma_{exp}} \right)^2}{n - m}$$

The denominator is the number of degrees of freedom ν , given by the number of datapoints n minus the number of fitting parameters m . There are four fitting parameters (thickness z , tilts α_0 and β_0 , and flake bending γ) and roughly 76 datapoints (77 total Bragg beams, minus the central spot and any spots clipped by the edges of the CCD screen). In principle, an accurate fit should yield $\chi_v^2 \sim 1$. A fit with $\chi_v^2 > 1$ indicates that the model is not fully capturing the data, or that the experimental error is underestimated. Our best fits are in the range of $\chi_v^2 \sim 2 - 5$. Given the complexity of dynamical electron scattering, this is a reasonable result. It is likely that consideration of effects such as diffuse thermal scattering or partial coherence of the beam would reduce the χ_v^2 value. Nevertheless, the current model is sufficient to differentiate the different stacking orders.

2. The authors show order/disorder via cross-sectional STEM in few-layer MoTe₂, but how does this compare to the bulk crystal for low temperature? Does it have the same structural disorder problems to the T_d structure?

RESPONSE 3.2: We find two ways to interpret this question.

First interpretation – for bulk crystals, is there disordered stacking at low temperature? This question is addressed in ref. 33 of the main text. This is a single-crystal neutron diffraction study of the 1T' to T_d phase transition in a high-quality bulk MoTe₂ crystal. The study finds well-ordered 1T' stacking at 295 K, and well-ordered T_d stacking at 100 K. Thus, at low temperature, the bulk crystal enters an ordered T_d structure.

Second interpretation – for thin flakes of MoTe₂ which show disordered stacking at room temperature, is there also disordered stacking at low temperature? This question is addressed in Fig. 5 of the main text, and in lines 276 – 282. In short, thin flakes of MoTe₂ show disordered stacking across the entire studied temperature range, from 10 K up to 675 K. For thinner flakes (10 nm and less) we see no evidence of temperature-induced layer sliding, and these flakes possess disordered stacking which is temperature independent. For thicker flakes (10s of nm in thickness), upon cooling below 100 K, we observe cooling-induced layer sliding and an increase in local T_d-like structure. However, there is no transition to a fully ordered T_d state, even with cooling down to 17 K. Hence, there is persistent disorder even at low temperatures.

3. Have the authors checked CVD grown samples of 1T'-MoTe₂, where no exfoliation was taken? This would be a better way of ruling out any effects from the exfoliation process.

RESPONSE 3.3: We agree with the reviewer's comment; however, these measurements are not feasible as there are no CVD-grown octahedral MoTe₂ flakes that are thick (>> 4 layers) where we can observe the layer disorder. Thus, while we agree that the referee's suggestion is excellent, it is outside the scope of our manuscript. Referee 1 asked a similar question; Please see our Response 1.2 above, which lists the CVD growth papers and thicknesses of MoTe₂ obtained (Response Table 1).

4. What happens to the stacking order at low temperature when the exfoliated flakes are thinner than the average T_d domain size? Why was this study limited to looking at flakes of 7 nm or greater?

RESPONSE 3.4: For relatively thick flakes (10s of nm), we observe an increase in local T_d stacking at low temperature, with an estimated T_d domain thickness of 4 nm. Conversely, for flakes < 10 nm in thickness, our electronic transport data (as well as transport data from refs. 17 and 18 of the main text), indicate that there are no changes in layer stacking with temperature. Hence, for flakes < 10 nm in thickness, our data suggests that the layer stacking is completely temperature independent, and there is no preference for T_d stacking.

Our STEM approach requires flakes of uniform thickness and lateral dimensions of $10+ \mu\text{m}$, to enable two orthogonal FIB lift-outs. Finding uniform flakes $< 10 \text{ nm}$ in thickness and lateral dimensions of $10+ \mu\text{m}$ is rare. Thus, our study was limited to flakes 7 nm or greater. Further, layer disorder can be studied for MoTe_2 flakes that are much thicker than 4 layers. Monolayer and bilayer MoTe_2 do not have any distinct stacking arrangements. A 3-layer flake can be $1T'$ ($\downarrow\downarrow$ or $\uparrow\uparrow$) or T_d ($\downarrow\uparrow$ or $\uparrow\downarrow$), but alternative stacking arrangements are not possible. A 4-layer flake has $2^3 = 8$ possible stacking arrangements, 4 of which are ordered $1T'$ and T_d , and the other 4 arrangements are mixed / disordered. For thicker flakes, the number of ordered stacking arrangements remains fixed at 4 ($\downarrow\downarrow\downarrow\dots$, $\uparrow\uparrow\uparrow\dots$, $\downarrow\downarrow\uparrow\dots$, $\uparrow\downarrow\uparrow\dots$), but the number of mixed / disordered stacking arrangements grows exponentially. Hence, if disordered stacking is present, it is best studied in flakes $\gg 4$ layers thick.

5. Figure 6 is in a variety of ways, misleading and not necessary for the story of the paper. It is incredibly speculative and should not be in the main text. The figure suggests things like gradual boundaries and transitions back to an ordered T' state at higher temperatures for exfoliated flakes and is directly at odds with what is stated for the experimental observations in the paper (i.e., that no return from a disordered state at RT to an ordered $1T'$ structure is observed, regardless of increased temperature). The same can be said for the T_d part of this phase diagram. The diagram also indicates many additional points for which the data is not given in either the main text or the supplementary. In some cases, the diagram points to even thinner samples than anything shown in the figures (down to 6 nm for R vs T for instance). It is also not clear how the authors can say that the samples are in the mixed state versus the fully disordered state solely from the R vs T curves (as indicated on the figure).

RESPONSE 3.5: We accept the reviewer's comment, and we have removed Figure 6.

6. Similar problems exist for the authors' table 2. Many of the values are inconsistent with experimental observations (particularly for WTe_2). The authors admit this in the main manuscript. While it's important to check these calculations and see if anything can be gleaned from them, I'm not convinced that they are important for proving the main hypothesis of the manuscript and find them distracting from the main message instead. The discussion around the values calculated in table 2 could be much more concise and the table itself included in the supplementary instead.

RESPONSE 3.6: The point of Table 2 was to say that DFT calculations do not provide conclusive insights. We agree with the reviewer's comment that this can be distracting to readers, and we have moved Table 2 to the Supplementary Information.

7. On line 312, the authors state that Fig 1b is indicative of high quality MoTe_2 crystals. However, their residual resistivity ratios from Fig. 1b for the bulk crystal is just 10. This indicates an incredible amount of defects in the material, likely in-plane point defects.

The authors argue that this disorder does not impact their results, stating that they have explored crystals from a variety of providers and growth conditions. Again, none of these results are shown in the main manuscript or supplementary and it would be great to know what kind of comparisons the authors made between differing quality material.

RESPONSE 3.7: We thank the reviewer for this comment. We have studied thin exfoliated flakes from bulk crystals having RRRs in the range of 400 – 800. The STEM imaging clearly shows disordered stacking, consistent with our prior results. The data is shown in Supplementary Figure 11 and also copied below as Response Figure 4 (this is the same as Response Figure 3, but copied below for convenience). We have also added data in the Supplementary which compares diffraction from all three MoTe₂ crystal sources (Supplementary Figure 13).

Response Figure 4. **a** Example resistance versus temperature data for a crystal grown with the conditions described in refs. 30 and 33 of the main text. For this crystal, the RRR = 478. A separate crystal showed RRR = 797. **b** Optical image of a MoTe₂ flake with top and bottom hexagonal boron nitride encapsulation layers. Exfoliation and heterostructure assembly were performed in an Ar glovebox with O₂ and H₂O levels < 0.5 ppm, using MoTe₂ crystals as described in refs. 30 and 33 of the main text. **c** HAADF-STEM image of the same heterostructure, demonstrating the presence of layer stacking disorder.

8. Finally, I would like to note that the graphite encapsulation utilized by the authors is insufficient at preventing oxidation of few-layer, air-sensitive TMDs. H₂O/O₂ can still easily diffuse through the SiO₂/MoTe₂ interface and even under the graphite/MoTe₂ interface unless the TMD is fully encapsulated on all sides.

RESPONSE 3.8: We thank the reviewer for this comment. We have studied a thin MoTe₂ flake fully encapsulated in h-BN. The data is shown in Supplementary Figure 11, as well as Response Figure 4, above. The STEM imaging clearly shows disordered stacking.

REVIEWERS' COMMENTS

Reviewer #1 (Remarks to the Author):

I have read through the revised manuscript and the reply to the comments and found that the authors have appropriately answered the comments and satisfactorily revised the manuscript. I think that the answers to the other referees are reasonable and persuasive. Therefore, I recommend the publication in Nature Communications.

Reviewer #3 (Remarks to the Author):

The authors have not only adequately addressed my concerns with the manuscript, but have demonstrated great diligence in addressing the concerns of the other reviewers as well. I disagree with referee #2 that this should be published in a more specific journal, as the authors point out that this interlayer disorder may be far more reaching than just MoTe₂, with this paper laying the groundwork for observing and understanding the disorder. I recommend it be published as is.

Reviewer #1 (Remarks to the Author):

I have read through the revised manuscript and the reply to the comments and found that the authors have appropriately answered the comments and satisfactorily revised the manuscript. I think that the answers to the other referees are reasonable and persuasive. Therefore, I recommend the publication in Nature Communications.

Reviewer #3 (Remarks to the Author):

The authors have not only adequately addressed my concerns with the manuscript, but have demonstrated great diligence in addressing the concerns of the other reviewers as well. I disagree with referee #2 that this should be published in a more specific journal, as the authors point out that this interlayer disorder may be far more reaching than just MoTe₂, with this paper laying the groundwork for observing and understanding the disorder. I recommend it be published as is.

AUTHOR RESPONSE: We thank the Reviewers 1 and 3 for reading our manuscript and providing valuable feedback.